# Algorithm for using dual energy computed tomography to determine chemical composition: A feasibility study

**Dong Hyeok Choi[1,2,3], So Hyun Ahn**  **[4,5]☯\*, Kwangwoo Park[6]☯\*, Min Cheol Han[3], Jin Sung Kim[1,2,3]**

**1** Department of Medicine, Yonsei University College of Medicine, Seoul, Republic of Korea, **2** Medical Physics and Biomedical Engineering Lab (MPBEL), Yonsei University College of Medicine, Seoul, Republic of Korea, **3** Department of Radiation Oncology, Yonsei Cancer Center, Heavy Ion Therapy Research, Seoul, Republic of Korea, **4** Ewha Medical Research Institute, School of Medicine, Ewha Womans University, Seoul, Republic of Korea, **5** Department of Biomedical Engineering, Ewha Womans University College of Medicine, Seoul, Republic of Korea, **6** Department of Radiation Oncology, Yongin Severance Hospital, Yonsei University College of Medicine, Yongin, Republic of Korea

☯ These authors contributed equally to this work.
\* mpsohyun@ewha.ac.kr (SHA); kpark02@yuhs.ac (KP)

## Abstract

Using dual-energy computed tomography (CT), this study aims to develop an algorithm to identify the chemical constituents of an unknown material (compound or mixture) and improve the accuracy of material discrimination. The algorithm requires mass attenuation coefficients (MAC) that were obtained using a dual energy CT as an input, identifies the elemental composition, and then calculates its weight fraction. To evaluate the functionality of the developed algorithm, it was used to determine the chemical constituents for human tissues. Furthermore, the results were compared with those provided by the National Institute of Standards and Technology (NIST). We used dual energies 80/140 kVp for spectral CT scans, as inputs to the algorithm, in addition to a set of 50/80 and 80/100 keV for mono-energetic X-rays. The algorithm correctly determined the chemical constituent elements of unknown materials. Results were obtained for the fractional weights of each component for mono-energetic X-rays and spectral X-ray use. For mono-energetic X-rays, the differences were < 0.01% for compounds and 6.02% for mixture, respectively. For the spectral X-rays, the differences in 2.98% for compounds and 6.03% for mixtures, respectively. We developed an algorithm to determine the type and weight fraction of an element using the MAC of dual-energy CT. The algorithm can exclude the inherent uncertainty of SPR calculations and improve the accuracy of dose calculations in radiation therapy planning.

## Introduction

In terms of physical and radiobiological properties, proton and heavy ion therapy have multiple advantages over photon therapy and thus have received considerable

**Data availability statement:** All relevant data are within the manuscript and its Supporting Information files.

**Funding:** This study was financially supported by the Basic Science Research Program through the National Research Foundation of Korea (NRF) funded by the Ministry of Education in the form of a grant (NRF-2022R1I1A1A01071588) received by KP. This study was also financially supported by Yonsei University College of Medicine in the form of a faculty research grant (6-2022-0062) received by KP. This study was also financially supported by the National Research Foundation of Korea (NRF) funded by the Korea government (MSIT) in the form of a grant (RS-2023-00240003) received by SHA. This study was also financially supported by the SME R&D project for the Start-up & Grow stage company, Ministry of SMEs and Startups in the form of a grant (RS-2024-00426787) received by SHA.

**Competing interests:** The authors have declared that no competing interests exist.

attention in the field of radiation therapy [1]. The most distinguishing features of particle therapy are that the relative biological effectiveness (RBE) and linear energy transfer (LET) are sufficiently high that the absorbed dose is high in the tumor and decreases in the surrounding normal tissue [2]. However, efforts are required for reducing uncertainty in dose calculation; otherwise, the uncertainly delivered high dose to organs at risk (OARs) could cause side effects.

In general, the calculation of the physical dose of particle radiation in the treatment planning system (TPS) is based on the stopping power ratio (SPR) [3]. According to the Bethe–Bloch equation, electron density and mean excitation potential ($I_m$) of materials play an important role for calculating SPR [3].

Recently, considerable efforts have been made to reduce the uncertainty of dose calculation using dual-energy computed tomography (DECT) [4,5,6]. As an extension of photon dose calculation, the method using electron density and effective atomic number can be employed to that of particles [7–12]. Zhu et al. estimated SPR using effective atomic number ($Z_{eff}$) from DECT [7]. The difference reduced to 1.4% for DECT, compared to 5.7% for single-energy computed tomography (SECT), demonstrated that the method using DECT resulted in improved material discrimination. Jung et al [11]. obtained $Z_{eff}$ using two SECT scans that may be replaced with a single DECT. Sakata et al [12]. reported a study to improve accuracy by increasing the energy gap between two dual energy images using kV and MV images.

Furthermore, from DECT images, there are multiple studies of model-based SPR calculations. Yang et al. suggested a parameterized and approximated Bethe equation to calculate SPR by determining $Z_{eff}$ and $I_m$, both of which can be obtained from fitting the known material data [12]. Another method is known as the basis vector model wherein mass attenuation coefficient (MAC) can be spanned by those of two known materials as a linear combination [13]. By determining coefficients in the linear combination for multiple human tissues, the $Z_{eff}$ of the material was estimated and used to compute the SPR.

Studies on improving SPR accuracy using DECT have also been conducted in tissue characterization methods. There are three major tissue characterization methods using DECT: (1) segmentation, (2) parametrization, and (3) decomposition. The segmentation method makes a list of materials in advance, assigns them to pixels, and accordingly determines the elemental composition. This method has the disadvantage that it is insensitive to changes in elemental composition from patient to patient [7]. The parametrization method has a database on the composition of human tissue; moreover, it expresses the weight fraction of an element as a function such as Hounsfield Units (HUs), electron density, and $Z_{eff}$. If the HU difference between human tissues in the database is not large, this method may be sensitive to noise and artifacts [14]. The decomposition method assumes that the human body is composed of substances such as water, lipid, and protein; moreover, it estimates the weight fraction of each of these base elements from electron density, and $Z_{eff}$ obtained from dual- or multiple-energy images [15,16]. The abovementioned methods have achieved remarkable development for calculating SPR and absorbed dose

calculation in particle therapy; however, when calculating the mean excitation energy, there is inevitably uncertainty when using $Z_{eff}$ compared to using the atomic number of each element.

Unlike the method of determining $Z_{eff}$, we attempted to determine exact chemical composition of unknown materials using the dual-energy CT, which would enable application for calculating SPR. To establish the concept of the method to determine constituent elements, the theory and algorithm of computation is demonstrated. Moreover, the results for the estimation of the chemical composition of unknown materials and their fractional weights are presented.

## Materials and methods

### Prerequisite of the algorithm

**Built-in data: mass attenuation coefficients (MACs).** The algorithm is working with MACs as a function of atomic number for dual (low and high) energy in the look-up table, built-in data, with an atomic number of 1–20. Two sets of MACs used were made as built-in data: the value provided by National Institute of Standards and Technology (NIST) and the Geant4 simulation. The two sets of built-in data comprise MACs for a ranging from 1 keV to 6 MV. NIST built-in data provides MACs to four decimal places, but Geant4 built-in data provides MACs to ten decimal places. Geant4 built-in data is expected to have the advantage of deriving additional accurate results when a CT image comprising multiple voxels is applied to the algorithm.

**Geant4 modeling.** For the MC simulation using Geant4, we set the model as follows (Fig 1):

• A point source of photon

• Narrow beam

• Number of photons: $10^8$

• Cylindrical gamma detector: radius of 0.1 mm $\times$ height of 0.1 mm

• Box-shaped material: $5 \times 5 \times 1$ cm$^3$

For using DECT, the American Association of Physicists in Medicine (AAPM) Task Group 291 Report (TG 291) suggested that the peak energies of spectral X-ray are 80 and 140 kVp to be sufficiently different to take advantage of the energy-dependent nature of MAC [17]. In this study, CT images based both on full energy spectra simulations and

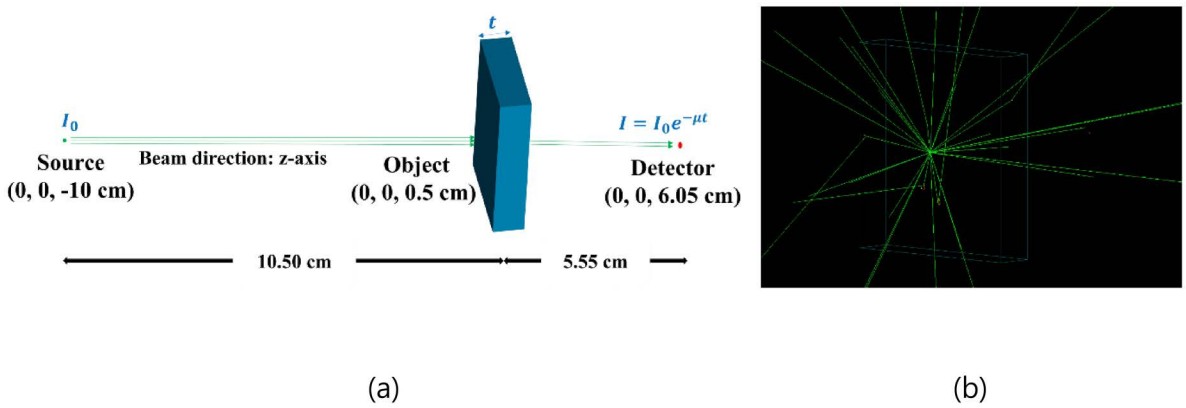

(a)                                                                 (b)

**Fig 1. (a) Geant4 modeling for calculating mass attenuation coefficients.** The blue box is the object for obtaining mass attenuation coefficient and has a thickness of 1 cm. (b) The tracks of X-rays in a simulation.

mono-energic beams were simulated. The peak energies of the energy spectra are 80/140 kVp, whereas the mono-energies are 50/80 and 80/100 keV, respectively. To confirm the built-in data, we compared the result of our simulations with NIST data for MACs for atomic numbers from 1 to 20.

**Identifying representative energy for spectral X-rays**

In principle, applying the algorithm for identifying chemical constituent from spectral energy, we should integrate over the complete energy spectrum. However, to save computational time and load, using mono energy representing the energy spectrum is considerably more efficient as long as the uncertainty remains sufficiently small. For the easy and efficient application for clinical use, the representative energy was defined as the energy at which the MAC of specific material (i.e., $H_2O$) was the same as when integrated over the full energy spectrum. In this study, to confirm the feasibility for clinical use, representative energy was determined using MC simulation rather than measurement. The MAC obtained by performing Geant4 simulation again with the representative energy obtained in this way was compared and evaluated with the MAC obtained by the Geant4 simulation for the full energy spectrum.

**Theory**

This algorithm can identify both the atomic number of the elements and their weight fractions for unknown mixtures or compounds. The algorithm can be defined as follows. The MAC of a material $x$ comprising $N$ chemical constituents can be spanned as a linear combination:

$$\left(\frac{\mu}{\rho}\right)_x = \sum_{i=1}^{N} w_i \left(\frac{\mu}{\rho}\right)_i, \text{ with fractional weight, } w_i = \frac{n_i A_i}{\sum_j n_j A_j} \tag{1}$$

where $w_i$ and $A_i$ are the weight fraction and atomic weight of element $i$ in the unknown material, $n_i$ is the number of atoms in molar amount, $x$. Assuming that MAC is obtained for a material composed of two unknown elements using two energies of high and low, Eq 1 can be expressed as Eq 2.

$$\mu_{x,L} = w \cdot \mu_{1,L} + (1 - w) \cdot \mu_{2,L}$$

$$\mu_{x,H} = w \cdot \mu_{1,H} + (1 - w) \cdot \mu_{2,H} \tag{2}$$

For simplicity, $\left(\frac{\mu}{\rho}\right)$ set to be $\mu$. Moreover, $w$ should sum to 1 over the elements; therefore, $w$ should reside in the range of [0,1]. Solving the system of Equations of Eq 2 for $w$ yields the following:

$$0 \leq w = \frac{\Delta\mu_2 - \Delta\mu_x}{\Delta\mu_2 - \Delta\mu_1} \leq 1 \tag{3}$$

where the mass difference in attenuation coefficient is $\Delta\mu = \mu_L - \mu_H$. Substituting Eq 3 in Eq 2 provides

$$\alpha \cdot \Delta\mu_2 + \beta \cdot \mu_{2,L} + \gamma = 0, \text{ with } \begin{cases} \alpha = \mu_{x,L} - \mu_{1,L} \\ \beta = \Delta\mu_1 - \Delta\mu_x \\ \gamma = \Delta\mu_x \cdot \mu_{1,L} - \Delta\mu_1 \cdot \mu_{x,L} \end{cases} \tag{4}$$

We solve Eq 4 for the MACs of material 2 ($\mu_{2L,H}$) with given unknown material's $\mu_{x,L,H}$ and test chemical constituent's $\mu_{1,L,H}$. Note that $\alpha$, $\beta$, and $\gamma$ are constants for a given input material $x$. The algorithm starts by assuming that hydrogen,

atomic number 1, is material 1. The overall process of determining atomic numbers and weight fractions based on the MAC information is depicted in Fig 2.

## Minimization

The system of equations, Eq 2, is not solvable because there are three unknowns with two equations. Thus, we should conditionally solve equations. Let us define Eq 4 to be an objective function ($f$):

$$f(z) = (\alpha \cdot \Delta \mu_2 + \beta \cdot \mu_2 + \gamma)^2 \qquad (5)$$

Where z denotes the atomic number. The reason for using a squared term in Eq 5, unlike Eq 4, is to prevent negative values and to facilitate the identification of the minimum value. Ideally, as expressed in Eq 4, $f(z)$ should vanish. However, the noise in the image and uncertainty in measurement, $f(z)$ would not vanish but become close to 0. Therefore, to determine material 2, $f(z)$ might be an objective function for minimization. For robustness against noise, we consider multiple local extrema during the process of minimization (green arrows in Fig 4). Note that the determination of trial material 1 is under the condition that material 2 exists.

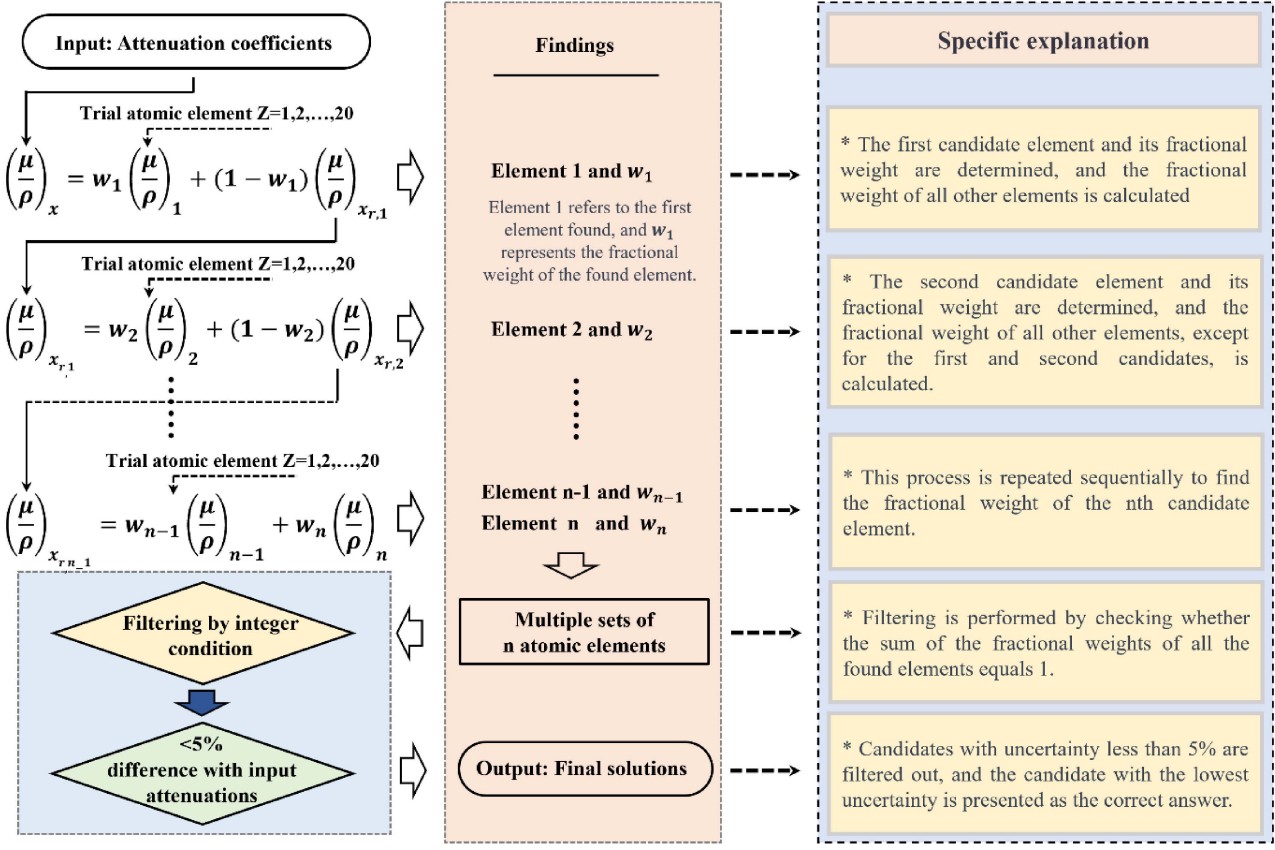

**Fig 2. The method for determining the chemical constituents of unknown materials in the algorithm.** $x$ is unknown material, $x_{r1}$ is all elements in the unknown material except the first element, $n$ is total number of elements in the unknown material.

## Integer condition

The minimization of objective function yields multiple solutions with fractional weights as in Eq 3 and chemical constituent. Using these results, the chemical expression can be determined from each number of atoms. In general, the expressed number of each constituent is

$$n_i = \frac{\mu_i}{\left(1 - \sum_j^{N-1} \mu_j\right) A_i} n_N \in I$$

(6)

where $I$ is a positive integer group with the material comprising N constituents ($i = 1, 2, \ldots, N-1, N$), $n_i$ is number of $i^{th}$ constituent and $n_N$ is the number of $N^{th}$ constituent. The program determines the termination time by designating the current number of elements in the algorithm. Moreover, there is an integer condition for the atomic number (Z); however, if it is found in ±0.1 of an arbitrary atomic number (Z) and it is terminated. This integer condition is extremely strict and rejects multiple solutions that do not meet most conditions in identifying compound materials. For this computation, based on these integer numbers of chemical constituents, MACs (A) are recalculated based on the material composition in the algorithm to compare the input MACs (B). If differences between (A) and (B) falls within the acceptance range (e.g., 5%), it is output as a result, otherwise it is excluded. The algorithm allows you to set the acceptance range for uncertainty from 1% to 10%.

Using the following mechanism, the algorithm identifies both atomic number and fractional weight; The identified atoms are numbered from 1 to N in descending order of fractional weight. That is, the atom with the highest fractional weight becomes $1^{st}$ element, and the atom with the lowest fractional weight becomes $N^{th}$ element.

## Algorithm execution

For Fig 3, the overall flowchart and process in the algorithm are demonstrated where we considered the application to the clinical use by replacing input MAC converted from the DECT image. To execute the algorithm, two essential inputs are required: monoenergies and MACs. If the spectrum energy is utilized, representative monoenergies are identified, as shown in the blue box of Fig 3, and employed as input data for the algorithm. Subsequently, utilizing a lookup table of MACs and atomic numbers, the algorithm obtains the respective MACs for low energy and high energy. Based on the information provided in Methods section, the algorithm then proceeds to determine the element and fractional weight.

## Example of identifying chemical constituents

The algorithm sequentially tests the atomic number starting with the atomic number 1, i.e., it is a method of identifying the next element assuming that the $x^{th}$ atomic number has been reported; if all of the specific $x_i$ results fail, the $x_{i+1}$ test is performed.

As an example of identifying the atomic number of an unknown material through the formulas mentioned in 2.2.1 and 2.2.2, the unknown material's MACs are 0.2074 and 0.1641 with mono energies of 50 and 100 keV, which were inputted for the algorithm. Fig 4 shows the process of identifying the next atomic number with the trial test element of C (Z = 6) after a first step to identify H (Z = 1). The objective function provides multiple solutions with fractional weight as local extrema in a log-scaled graph where the value of 8.00 among local extrema (green arrows in Fig 4) was determined as a third element; however, other singularities are rejected according to the condition that the fractional weight of each element must be within [0,1] (blue curve). Therefore, H-C-N and H-C-O were the possible combinations of chemical constituents. By applying Eq 6, the number of constituents can be obtained, followed by the determination of chemical formulae. Finally, the algorithm compares two MACs: acquired from chemical formulas using Eq 1 and provided as an input to determine the chemical constituent of an unknown material as polymethyl methacrylate ($C_5O_2H_8$) which has less uncertainty.

The algorithm was tested on five compounds that combine C, H, O elements and nine mixtures of the body tissue materials mentioned [18,19]. In the case of body tissue materials, the fractional weight of some elements was slightly modified so that the sum of the fractional weights was 100%.

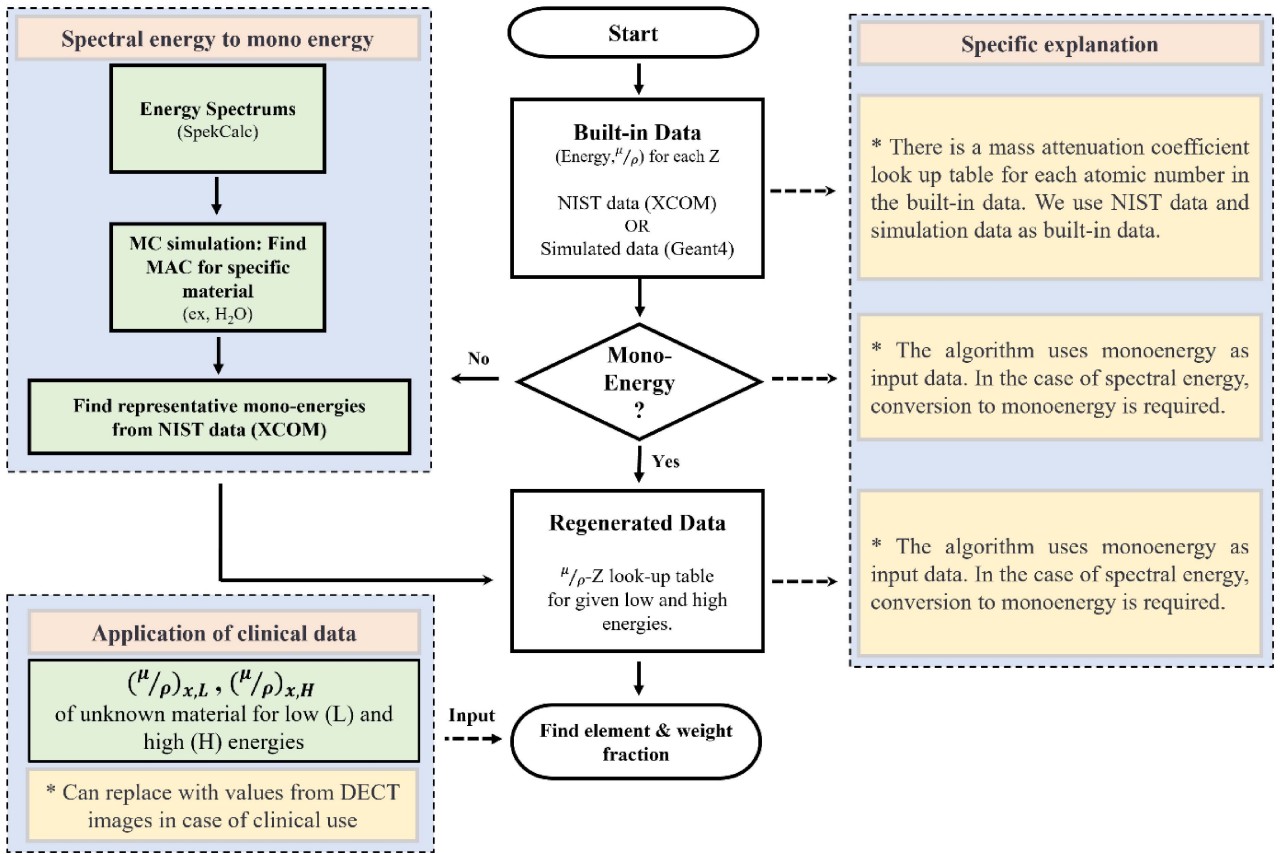

**Fig 3. Depicts the schematic of the algorithm.** The clinical use was considered and shown in the dashed line box as a replacement.

## Results

### Prerequisite of the algorithm

**Validation of Geant4 simulation.** As shown in Fig 5, MACs obtained using Geant4 simulation for every element were compared using NIST data and demonstrated that resultant percentage differences reside in the range of −1.30%–0.28% and −0.79%–0.08% for 50 and 80 keV.

**Determination of representative energy.** Validating the method for the representative energy of spectral X-ray, the MAC of $H_2O$, PMMA, and cortical bone (ICRP) [17] obtained through the energy spectrum comparing it with that from representative energy (Table 1). Consequently, the mono energies representing 80 and 140 kVp were 40.13–44.29 and 49.50–53.31 keV for three materials. Furthermore, the difference in the absolute values of the MACs of spectrum energy ($\mu/\rho_{spec\ E}$) and representative energy ($\mu/\rho_{rep\ E}$) was 0.07–1.05%. To determine the representative energy for water ($H_2O$) from the 140 kVp X-ray spectrum, we employed SpekCalc software followed by Geant4 Monte Carlo simulation validation. The detailed methodology for this energy calculation process is provided in the Supporting Information (S1 File).

$$(C) = \left| \frac{(B) - (A)}{(A)} \right| \times 100(\%)$$

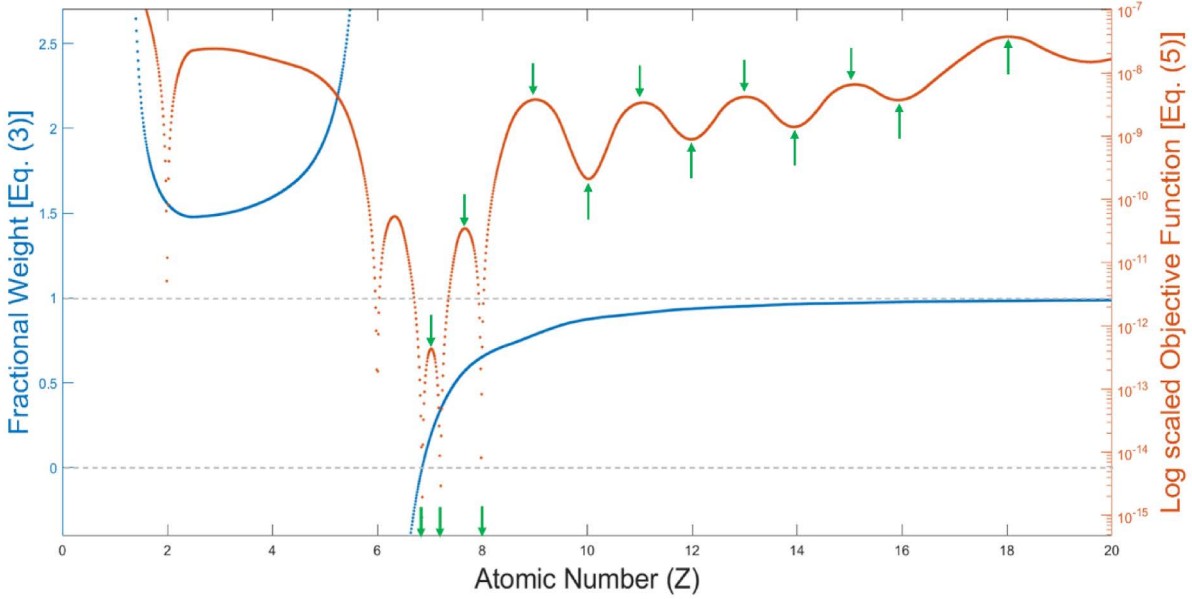

**Fig 4. Test of polymethyl methacrylate ($C_5O_2H_8$) through algorithm: Curves of both objective function and fractional weight in the process of identifying third chemical constituent after H was assumed, i.e., step 2 in Figure 2.** The trial element, C (Z = 6), brought objective function to the minimal singularity at Z = 8.

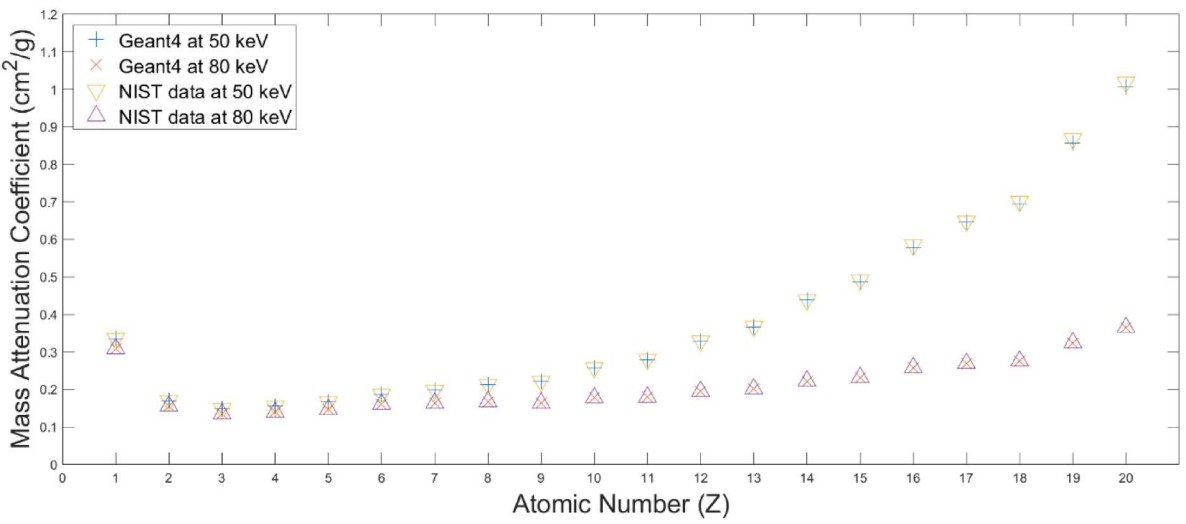

**Fig 5. Comparison of mass attenuation coefficient calculated by Geant4 simulation and acquired from NIST data.** Inverted triangle (yellow) and triangle (purple) mean the NIST data for 50 and 80 keV, respectively. Cross (blue) and X (red) mean the data by Geant4 simulation at 50 and 80 keV, respectively.

### Identifying chemical constituents

**Compounds.** The results of the unknown material's chemical constituents with fractional weights were obtained by comparing the values calculated from NIST data using Eq 1. Consequently, the percentage differences in fractional weights of chemical constituent for five compounds were in <0.01% and 2.98% for the material comprising two and three

**Table 1. Comparison of calculated mass attenuations using energy spectrum and their representative energies at $H_2O$, PMMA, and cortical bone (ICRP).**

| Material | Density (g/cm³) | Spectral energy | | Representative energy | | %diff (C) |
|---|---|---|---|---|---|---|
| | | Energy (kVp) | $\mu/\rho_{specE}$ (cm²/g) (A) | Energy (keV) | $\mu/\rho specE$ (cm²/g) (B) | |
| $H_2O$ | 1.00 | 80 | 0.2689 | 41.70 | 0.2682 | 0.07 |
| | | 140 | 0.2274 | 49.50 | 0.2297 | 0.22 |
| PMMA | 1.18 | 80 | 0.2344 | 40.13 | 0.2342 | 0.09 |
| | | 140 | 0.2057 | 50.80 | 0.2057 | < 0.01 |
| Cortical bone (ICRP) | 1.85 | 80 | 0.5408 | 44.29 | 0.5303 | 1.05 |
| | | 140 | 0.3775 | 53.31 | 0.3772 | 0.03 |

constituents, respectively. These results demonstrate that the algorithm for identifying the chemical constituents works quite accurately for the material comprising two or three constituents with H, C, and O (Table 2).

## Mixtures

Nine human tissue (mixture) materials were tested and the results are shown in Table 3. It was confirmed that the %difference in the case of number of constituents was 4, 5 and 6 were 0.340–5.449%, 1.410–5.188%, and 3.693–6.027%, respectively. It can be seen that the higher the number of constituents, the higher the %difference overall. The reasons for this will be discussed in more detail in Discussion section.

## Discussion

The originality of our algorithm is that it can find the constituent elements of an unknown material consisting of two or more elements and the fractional weight of those elements using only two CT energy spectra. Although the algorithm we developed does not require calculating $Z_{eff}$ as an approximation, it is considered a useful result, but several considerations were made about the problems encountered during performing of this study and are described below.

### Consideration on the improvement of the accuracy

We made careful efforts to improve the accuracy in the overall process of algorithm development. One of them was the effort made in the process of selecting built-in data to be used in the process of identifying chemical constituents. Rather than using the MAC provided by NIST, we used that obtained by Geant4 simulation as built-in data because NIST only provides MACs with four significant figures. Moreover, the resulting truncation error possibly acts as a factor of uncertainties. Because MACs obtained by Geant4 simulation might not completely remove uncertainty, the condition that the atomic number of an element is always quantized to an integer was used in the algorithm.

Currently, the suggested algorithm can identify chemical constituents for mixtures or compounds with two elements with extremely high accuracy (Table 2). However, the confirmation that the accuracy of the result was somewhat lower for mixtures or compounds with more than three elements. The accuracy decreases as the number of elements forming the material increases can be understood by considering error propagation. Fig 2 shows the algorithm sequentially identifies the atomic number and weight fraction step by step. In this process of identifying the 1, 2, …, and $(n)^{th}$ element, the MAC from the first to the $(n-2)^{th}$ elements are used as inputs for the subsequent step. Therefore, at each step, the resultant atomic number $Z_{step}$ has the uncertainty propagated via the objective function ($f$) (Eq 5) [20]:

$$\sigma_{Z_{step\ 1}} = \sqrt{\left(\frac{\partial f}{\partial \mu_x}\sigma_{\mu_x}\right)^2 + \left(\frac{\partial f}{\partial \mu_1}\sigma_{\mu_1}\right)^2}$$

(7)

**Table 2. Results of using the algorithm to find the chemical constituents for compounds using dual energy and mass attenuation coefficient.**

| Human Body Material | Dual Energy Type | Number of Constituent | H | C | N | O | P | Ca | Max. Diff. |
|---|---|---|---|---|---|---|---|---|---|
| $H_2O$ | Ref. | 2 | 11.20% | N/A | N/A | 88.80% | N/A | N/A | |
| | Mono. I | | 11.191% (−0.009%) | | | 88.809% (0.009%) | | | 0.009% |
| | Mono. II | | 11.191% (−0.009%) | | | 88.809% (0.009%) | | | 0.009% |
| | Spectral I | | 11.191% (−0.009%) | | | 88.809% (0.009%) | | | 0.009% |
| | Spectral II | | 11.191% (−0.009%) | | | 88.809% (0.009%) | | | 0.009% |
| | Spectral III | | 11.191% (−0.009%) | | | 88.809% (0.009%) | | | 0.009% |
| $C_2H_4$ | Ref. | 2 | 14.37% | 85.63% | N/A | N/A | N/A | N/A | 0.002% |
| | Mono. I | | 14.372% (0.002%) | 85.628% (−0.002%) | | | | | 0.002% |
| | Mono. II | | 14.372% (0.002%) | 85.628% (−0.002%) | | | | | 0.002% |
| | Spectral I | | 14.372% (0.002%) | 85.628% (−0.002%) | | | | | 0.002% |
| | Spectral II | | 14.372% (0.002%) | 85.628% (−0.002%) | | | | | 0.002% |
| | Spectral III | | 14.372% (0.002%) | 85.628% (−0.002%) | | | | | 0.002% |
| $C_3H_8$ | Ref. | 2 | 7.74% | 92.26% | N/A | N/A | N/A | N/A | 0.003% |
| | Mono. I | | 7.743% (0.003%) | 92.258% (−0.002%) | | | | | 0.003% |
| | Mono. II | | 7.743% (0.003%) | 92.258% (−0.002%) | | | | | 0.003% |
| | Spectral I | | 7.743% (0.003%) | 92.258% (−0.002%) | | | | | 0.003% |
| | Spectral II | | 7.743% (0.003%) | 92.258% (−0.002%) | | | | | 0.003% |
| | Spectral III | | 7.743% (0.003%) | 92.258% (−0.002%) | | | | | 0.003% |
| $C_5O_2H_8$ | Ref. | 3 | 8.05% | 59.98% | N/A | 31.96% | N/A | N/A | 0.005% |
| | Mono. I | | 8.055% (0.005%) | 59.985% (0.005%) | | 31.961% (0.001%) | | | 0.005% |
| | Mono. II | | 8.055% (0.005%) | 59.985% (0.005%) | | 31.961% (0.001%) | | | 0.005% |
| | Spectral I | | 8.055% (0.005%) | 59.985% (0.005%) | | 31.961% (0.001%) | | | 0.005% |
| | Spectral II | | 8.055% (0.005%) | 59.985% (0.005%) | | 31.961% (0.001%) | | | 0.005% |
| | Spectral III | | 8.055% (0.005%) | 59.985% (0.005%) | | 31.961% (0.001%) | | | 0.005% |
| $C_{10}H_8O_4$ | Ref. | 3 | 4.20% | 62.50% | N/A | 33.30% | N/A | N/A | 0.004% |
| | Mono. I | | 4.196% (−0.004%) | 62.502% (0.002%) | | 33.302% (0.002%) | | | 0.004% |
| | Mono. II | | 4.196% (−0.004%) | 62.502% (0.002%) | | 33.302% (0.002%) | | | 0.004% |
| | Spectral I | | 4.196% (−0.004%) | 62.502% (0.002%) | | 33.302% (0.002%) | | | 0.004% |
| | Spectral II | | 3.980% (−0.220%) | 62.000% (−0.491%) | | 34.011% (0.711%) | | | 0.711% |
| | Spectral III | | 4.640% (0.440%) | 65.039% (2.539%) | | 30.322% (−2.978%) | | | 2.978% |

Mono I: 50/80 keV; Mono II: 80/100 keV; Spectral I: Representative energy for 80/140 kVp at $H_2O$; Spectral II: Representative energy for 80/140 kVp at PMMA; Spectral III: Representative energy for 80/140 kVp at cortical bone.

where $\left(\frac{\mu}{\rho}\right)_x \equiv \mu_x$ and $\left(\frac{\mu}{\rho}\right)_1 \equiv \mu_1$ are considered as variables because the MAC of unknown material, $\left(\frac{\mu}{\rho}\right)_x$, as an input value has the uncertainty that could be from measurement or reconstruction of CT scan. Furthermore, the MAC of trial element $\left(\frac{\mu}{\rho}\right)_1$ might have truncation errors in the computation. With increase in steps, the uncertainty would increase.

$$\sigma_{tot} = \sqrt{\left(\sigma_{z_{step\ 1}}\right)^2 + \left(\sigma_{z_{step\ 2}}\right)^2 + \cdots + \left(\sigma_{z_{step\ n}}\right)^2} \qquad (8)$$

Therefore, as the number of constituent elements increase, the uncertainty inevitably increases as the uncertainty propagates. In future, we expect to determine the chemical constituents for a material of >10 elements using the developed algorithm. Therefore, in order to improve accuracy, it will be more essential to use the MAC calculated by Geant4 rather than the MAC of NIST in the future.

**Table 3. Results of using the algorithm to find the chemical constituents for mixtures using dual energy and mass attenuation coefficient.**

| Human Body Material | Dual Energy Type | Number of Constituent | Fractional Weight Per Chemical Constituent | | | | | | Max. Diff. |
|---|---|---|---|---|---|---|---|---|---|
| | | | H | C | N | O | P | Ca | |
| Adipose Tissue | Ref. | 4 | 11.40% | 60.00% | 0.70% | 27.90% | N/A | N/A | |
| | Mono. I. | | 11.270% (−0.130%) | 58.608% (−1.392%) | 0.301% (−0.399%) | 29.821% (1.921%) | | | 1.921% |
| | Mono. II | | 11.299% (−0.101%) | 59.927% (−0.073%) | 0.887% (0.187%) | 27.887% (−0.013%) | | | 0.187% |
| | Spectral I. | | 11.318% (−0.082%) | 60.758% (0.758%) | 0.887% (0.187%) | 27.037% (−0.863%) | | | 0.863% |
| | Spectral II. | | 11.310% (−0.091%) | 60.729% (0.729%) | 0.887% (0.187%) | 27.074% (−0.826%) | | | 0.826% |
| | Spectral III. | | 11.304% (−0.096%) | 60.634% (0.634%) | 0.887% (0.187%) | 27.175% (−0.725%) | | | 0.725% |
| Yellow Marrow | Ref. | 4 | 11.50% | 64.60% | 0.70% | 23.20% | N/A | N/A | |
| | Mono. I. | | 11.184% (−0.316%) | 66.069% (1.469%) | 0.227% (−0.473%) | 22.519% (−0.681%) | | | 1.469% |
| | Mono. II | | 12.026% (0.526%) | 65.147% (0.547%) | 0.349% (−0.351%) | 22.479% (−0.721%) | | | 0.721% |
| | Spectral I. | | 1.241% (−0.259%) | 66.723% (2.123%) | 0.888% (0.188%) | 21.148% (−2.052%) | | | 2.123% |
| | Spectral II. | | 11.249% (−0.251%) | 66.730% (2.130%) | 0.888% (0.188%) | 21.133% (−2.067%) | | | 2.130% |
| | Spectral III. | | 11.245% (−0.255%) | 59.151% (−5.449%) | 0.888% (0.188%) | 28.717% (5.517%) | | | 5.449% |
| Lymph | Ref. | 4 | 10.90% | 4.10% | 1.10% | 83.90% | N/A | N/A | |
| | Mono. I. | | 10.543% (−0.357%) | 3.256% (−0.844%) | 0.895% (−0.205%) | 85.273% (1.373%) | | | 1.373% |
| | Mono. II | | 10.560% (−0.340%) | 4.297% (0.197%) | 0.894% (−0.206%) | 84.187% (0.287%) | | | 0.340% |
| | Spectral I. | | 10.497% (−0.403%) | 0.895% (−3.205%) | 0.886% (−0.214%) | 87.722% (3.822%) | | | 3.822% |
| | Spectral II. | | 10.505% (−0.395%) | 0.895% (−3.205%) | 0.886% (−0.214%) | 87.714% (3.814%) | | | 3.814% |
| | Spectral III. | | 10.550% (−0.350%) | 0.894% (−3.206%) | 0.886% (−0.214%) | 87.670% (3.770%) | | | 3.770% |
| Human Body Material | Dual Energy Type | Number of Constituent | Fractional Weight Per Chemical Constituent | | | | | | Max. Diff. |
| | | | H | C | N | O | P | Ca | |
| Mammary Gland | Ref. | 5 | 10.90% | 50.80% | 2.30% | 35.90% | 0.10% | N/A | |
| | Mono. I. | | 10.818% (−0.082%) | 50.616% (−0.184%) | 0.386% (−1.914%) | 37.799% (1.899%) | 0.382% (0.282%) | | 1.914% |
| | Mono. II | | 11.030% (0.130%) | 51.640% (0.840%) | 0.890% (−1.410%) | 35.919% (0.019%) | 0.522% (0.422%) | | 1.410% |
| | Spectral I. | | 10.803% (−0.097%) | 50.830% (0.030%) | 0.384% (−1.916%) | 37.352% (1.452%) | 0.631% (0.531%) | | 1.916% |
| | Spectral II. | | 10.811% (−0.089%) | 50.870% (0.070%) | 0.383% (−1.917%) | 37.308% (1.408%) | 0.627% (0.527%) | | 1.917% |
| | Spectral III. | | 10.877% (−0.023%) | 51.126% (0.326%) | 0.380% (−1.920%) | 37.003% (1.103%) | 0.614% (0.514%) | | 1.920% |
| Red Marrow | Ref. | 5 | 10.60% | 41.70% | 3.40% | 44.20% | 0.10% | N/A | |
| | Mono. I. | | 10.419% (−0.181%) | 43.659% (1.959%) | 0.896% (−2.504%) | 44.521% (0.321%) | 0.505% (0.405%) | | 2.504% |
| | Mono. II | | 10.658% (0.058%) | 42.654% (0.954%) | 0.893% (−2.507%) | 45.253% (1.053%) | 0.542% (0.442%) | | 2.507% |
| | Spectral I. | | 10.472% (−0.128%) | 40.882% (−0.818%) | 0.486% (−2.914%) | 46.840% (2.640%) | 1.320% (1.220%) | | 2.914% |
| | Spectral II. | | 10.472% (−0.128%) | 40.905% (−0.795%) | 0.486% (−2.914%) | 46.818% (2.618%) | 1.319% (1.219%) | | 2.914% |
| | Spectral III. | | 10.493% (−0.107%) | 41.044% (−0.656%) | 0.485% (−2.915%) | 46.663% (2.463%) | 1.315% (1.215%) | | 2.915% |
| Prostate | Ref. | 5 | 10.60% | 9.00% | 2.50% | 77.80% | 0.10% | N/A | |
| | Mono. I. | | 10.097% (−0.503%) | 6.152% (−2.848%) | 0.899% (−1.601%) | 82.789% (4.989%) | 0.062% (−0.038%) | | 4.989% |
| | Mono. II | | 10.127% (−0.473%) | 5.904% (−3.096%) | 0.899% (−1.601%) | 82.988% (5.188%) | 0.083% (−0.017%) | | 5.188% |
| | Spectral I. | | 10.112% (−0.488%) | 6.237% (−2.763%) | 0.899% (−1.601%) | 82.689% (4.889%) | 0.063% (−0.037%) | | 4.899% |
| | Spectral II. | | 10.108% (−0.492%) | 6.227% (−2.773%) | 0.899% (−1.601%) | 82.703% (4.903%) | 0.063% (−0.037%) | | 4.903% |
| | Spectral III. | | 10.119% (−0.482%) | 6.202% (−2.798%) | 0.899% (−1.601%) | 82.719% (4.918%) | 0.063% (−0.037%) | | 4.918% |

*(Continued)*

**Table 3.** (Continued)

| Human Body Material | Dual Energy Type | Number of Constituent | Fractional Weight Per Chemical Constituent | | | | | | Max. Diff. |
|---|---|---|---|---|---|---|---|---|---|
| | | | H | C | N | O | P | Ca | |
| Sternum | Ref. | 6 | 7.80% | 31.80% | 3.70% | 44.10% | 4.00% | 8.60% | |
| | Mono. I. | | 11.040% (3.240%) | 35.493% (3.693%) | 0.708% (−2.992%) | 41.426% (−2.674%) | 0.528% (−3.472%) | 10.807% (2.207%) | 3.693% |
| | Mono. II | | 10.839% (3.039%) | 35.596% (3.796%) | 0.717% (−2.983%) | 41.726% (−2.374%) | 0.528% (−3.472%) | 10.594% (1.994%) | 3.796% |
| | Spectral I. | | 11.801% (4.001%) | 35.191% (3.391%) | 0.700% (−3.000%) | 40.680% (−3.420%) | 0.523% (−3.477%) | 11.105% (2.505%) | 4.001% |
| | Spectral II. | | 11.741% (3.941%) | 35.247% (3.447%) | 0.700% (−3.000%) | 40.705% (−3.395%) | 0.523% (−3.477%) | 11.084% (2.484%) | 3.941% |
| | Spectral III. | | 11.570% (3.770%) | 35.401% (3.601%) | 0.693% (−3.007%) | 40.815% (−3.285%) | 0.523% (−3.477%) | 10.998% (2.398%) | 3.770% |
| Femur (Total Bone) | Ref. | 6 | 6.40% | 33.40% | 2.90% | 36.30% | 6.60% | 14.40% | |
| | Mono. I. | | 8.195% (1.795%) | 32.873% (−0.527%) | 0.589% (−2.311%) | 41.129% (4.829%) | 0.583% (−6.017%) | 16.630% (2.230%) | 6.017% |
| | Mono. II | | 8.163% (1.763%) | 33.099% (−0.301%) | 0.587% (−2.313%) | 40.838% (4.538%) | 0.582% (−6.018%) | 16.732% (2.332%) | 6.018% |
| | Spectral I. | | 9.533% (3.133%) | 32.606% (−0.794%) | 0.579% (−2.321%) | 39.974% (3.674%) | 0.573% (−6.027%) | 16.736% (2.336%) | 6.027% |
| | Spectral II. | | 9.420% (3.020%) | 32.651% (−0.749%) | 0.579% (−2.321%) | 40.045% (3.745%) | 0.573% (−6.027%) | 16.731% (2.331%) | 6.027% |
| | Spectral III. | | 9.090% (2.690%) | 32.722% (−0.678%) | 0.582% (−2.318%) | 40.317% (4.017%) | 0.576% (−6.024%) | 16.713% (2.313%) | 6.024% |
| Ribs (2nd-6th) | Ref. | 6 | 6.50% | 26.50% | 3.90% | 43.90% | 6.00% | 13.20% | |
| | Mono. I. | | 7.335% (0.835%) | 29.339% (2.839%) | 8.383% (4.483%) | 38.800% (−5.100%) | 0.549% (−5.451%) | 15.594% (2.394%) | 5.451% |
| | Mono. II | | 7.238% (0.738%) | 28.614% (2.114%) | 9.079% (5.179%) | 38.757% (−5.143%) | 0.551% (−5.449%) | 15.761% (2.561%) | 5.449% |
| | Spectral I. | | 8.526% (2.026%) | 28.197% (1.697%) | 0.633% (−3.267%) | 46.610% (2.710%) | 0.626% (−5.374%) | 15.407% (2.207%) | 5.374% |
| | Spectral II. | | 8.439% (1.939%) | 28.368% (1.868%) | 0.632% (−3.268%) | 46.531% (2.631%) | 0.626% (−5.374%) | 15.405% (2.205%) | 5.374% |
| | Spectral III. | | 8.197% (1.697%) | 28.858% (2.358%) | 8.734% (4.834%) | 38.014% (−5.886%) | 0.542% (−5.458%) | 15.654% (2.454%) | 5.886% |

Mono I: 50/80 keV; Mono II: 80/100 keV; Spectral I: Representative energy for 80/140 kVp at $H_2O$; Spectral II: Representative energy for 80/140 kVp at PMMA; Spectral III: Representative energy for 80/140 kVp at cortical bone.

## Consideration on the improvement of calculation time

The suggested algorithm takes about 0.5 to 10 s for material with two and four elements, respectively (Intel i7-10700k CPU, 32 GB RAM). For application to medical images having about 100 slices with $512 \times 512$ voxels, it is necessary to improve the calculation time in terms of hardware and software. GPU-based programming would be appropriate; moreover, there will be another approach to impose certain restrictions for identifying an element. For example, it could be sufficient to use 10 chemical elements for medical CT scans. Alternatively, we can reduce the number of voxels by skipping the calculation of nearby and similar HU-valued voxels.

## Evaluating the effect of noise

To extend DECT research to the clinical field, it is essential to consider noise. Other researchers also considered the case in which noise is included in the image. Hünemohr et al. predicted the weight fraction by including a uniform Gaussian distributed noise with one standard deviation in electron density and $Z_{eff}$ [14]. They demonstrated that the mean standard deviation was 0.1% H, 9.9% C, 0.9% N, 10.3% O, 1.1% P, and 0.2% Ca. Lalonde and Bouchard reported a method of using a larger number of energy spectra compared to the number of principal components (PC) used in the principal components analysis (PCA) technique to overcome the effect of noise [15].

We investigated whether the developed algorithm could identify elemental composition well even when noise was present in the CT images reported by other authors; furthermore, we performed the noise test. The MAC is derived from the HU acquired via CT scans. In accordance with the directives outlined by the International Atomic Energy Agency (IAEA), it is recommended that the CT value's precision remains within a range of ± 20 HU relative to the manufacturer's recommended value [21]. Considering the allowable deviation of 20 HU, the MAC exhibits an error of about 2% in relation to water as the standard reference. With a prudent approach, we undertook error assessments, setting a conservative upper limit of 3%. Table 4 shows the results of the element composition and fractional weight of $H_2O$ reported in the algorithm when there is an error of 0.5%–5.0% in the MACs input to the algorithm. If an error of 3% or more is included in the MAC, it was confirmed that an error of 5% or more remains in the fractional weight calculated by the algorithm. Table 5 shows how much the stopping power ratio differs from the fractional weight difference calculated using the Bethe-Bloch formula [22]. Since our algorithm finds the ratio of constituent elements based on the difference in the MAC according to energy, research on finding an appropriate energy pair should be conducted in order to be robust to noise.

## Representative energy difference according to material

To test the uncertainty of using the representative energy obtained through $H_2O$ for other materials, the difference PMMA and cortical bone (ICRP) was compared to $H_2O$. Furthermore, representative energies for spectral X-ray, i.e., 80 and 140 kVp, were compared (Table 6). The representative energy differences between $H_2O$ and PMMA were 0.20 and 0.97 keV

**Table 4. The percentage difference between the weight fraction of the elements reported by the algorithm and ground truth when present in the error in the mass attenuation coefficients used as an input.**

| Noise (%) | Z = 1 | | Z = 8 | |
|---|---|---|---|---|
| | Fractional weight | % difference | Fractional weight | % difference |
| 0.0 | 0.111907 | 0.00 | 0.888093 | 0.00 |
| 0.5 | 0.120112 | 0.84 | 0.879888 | 0.84 |
| 1.0 | 0.128167 | 1.69 | 0.871833 | 1.69 |
| 1.5 | 0.136076 | 2.56 | 0.863924 | 2.56 |
| 2.0 | 0.143843 | 3.44 | 0.856157 | 3.44 |
| 2.5 | 0.158965 | 4.71 | 0.841035 | 4.71 |
| 3.0 | 0.158965 | 4.71 | 0.841035 | 4.71 |

**Table 5. SPR result as per 0% and 3% error of mass attenuation coefficient in H₂O.**

| Material | Z | L(eV) | A | Weight fraction | | SPR | | %difference |
|---|---|---|---|---|---|---|---|---|
| | | | | w/o deviation | 3% deviation | w/o deviation | 3% deviation | |
| H20 | 1 | 19.2 | 1.008 | 0.111898 | 0.161898 | 4.23 | 4.11 | 2.91 |
| | 2 | 95 | 16 | 0.838102 | 0.838102 | | | |

**Table 6. The results of representative energy (keV) for 80 and 140 kVp energies for H₂O, PMMA, and bone.**

| Material | Density (g/cm³) | Representative energy (keV) | |
|---|---|---|---|
| | | 80kvp | 140kVp |
| H2O | 1.00 | 39.93 | 49.84 |
| PMMA | 1.18 | 40.13 | 50.80 |
| Cortical bone (ICRP) | 1.85 | 44.29 | 53.31 |

at 80 and 140 kVp, respectively, and 4.17 and 2.51 keV between $H_2O$ and cortical bone. Consequently, for PMMA with a lower density compared to $H_2O$, the difference in representative energy was lower than that of the bone with a high density difference. To reduce the uncertainty stemming from the use of the representative energy, materials such as bone with a high density or lungs with a low density should be considered. In future, we plan to conduct research to obtain robustness representative energy for biomaterials.

### Statistical error of Geant4 simulation

Geant4 simulation was used to acquire built-in data and input data to perform algorithms. To acquire both data, 1E + 08 events were used for the simulation. To confirm whether data is statistically reliable, the number of events was changed and MACs was obtained ten times for each event. Table 7 shows the average, standard deviation, and relative error of MACs obtained as per the number of events. Relative error represents statistical precision because of fractions for the estimated mean; the equation is as follows.

$$Relative\ error = \left[ \frac{1}{N} \left( \frac{\overline{x^2}}{\overline{x}^2} - 1 \right) \right]^{1/2}$$

(9)

where N is the number of trials, $\overline{x^2}$ is the average of the squares of each trial result, and $\overline{x}$ is the average of trial results. The number of events used in this study (1E + 08) has a standard deviation of 5.68E-05, whereas the relative error value is < 0.01, which is sufficiently reliable.

**Table 7. Mean, standard deviation, and relative error results of mass attenuation coefficients as per the number of events used to obtain the mass attenuation coefficients for H₂O through Geant4 simulation.**

| I₀ | Mean | STD | Relative Error (%) |
|---|---|---|---|
| 1E + 03 | 0.187340 | 1.93E-02 | 13.88 |
| 1E + 04 | 0.182308 | 3.92E-03 | 6.26 |
| 1E + 05 | 0.184286 | 1.89E-03 | 4.35 |
| 1E + 06 | 0.183447 | 7.67E-04 | 2.77 |
| 1E + 07 | 0.183544 | 1.66E-04 | 1.29 |
| 1E + 08 | 0.183493 | 5.68E-05 | 0.75 |

## Clinical applicability

Several methods of deriving SPR from DECT data have been tested using experimental data [23–25]. The algorithm proposed in this study currently conducts image evaluation on substances consisting of three to six components, but it is necessary to decompose substances based on information on CT volume to be applied to clinical situations. In order to use the algorithm developed in this study for clinical use, the patient CT image will be partitioned by organ and the scope of finding solutions will be reduced by setting tolerance by organ. To this end, we would like to create a relief element and ratio table for each human organ to add built-in data, and provide the correct answer to the items with the smallest difference in the element ratio compared to the algorithm results. In addition, in order to improve the speed of algorithm calculation for CT information in the volume unit, the HU value of the volume divided by the long term will be replaced with an average value, and the algorithm will be executed by unifying it into one HU value for each long term. This will set a range with the same HU value for the same organ and optimize it to clarify the distinction from other organs.

This paper presents an initial study that primarily revolves around algorithm development and experimental validation. However, to acquire real-world results using clinical data, further steps involving thorough clinical verification and evaluation are necessary. Additionally, we are exploring additional research directions, including algorithm enhancement and data acquisition, to facilitate future clinical applications. It is crucial to conduct more research to obtain concrete outcomes from clinical data.

## Comparison with previous studies

Uncertainties arise when obtaining SPR from effective atomic number [7–13]. Zhu et al. pointed out that variations in the composition and density of human tissue due to factors like sex, age, and disease state can lead to uncertainties if the imaged material's composition does not match the material used for calibration. In this study, maximum errors of 12.8% and 2.2% were found for SECT and DECT, respectively.

Ohira et al. stated that the HU value for a given material with the same SPR can vary due to differences in electron densities and effective atomic numbers. Consequently, SPR prediction for radiation treatment planning using existing CT scanners may be inaccurate. The study reported uncertainties within −10% and 10% for $Z_{eff}$ in tissue substitutes with low or high atomic numbers.

Since $Z_{eff}$ is not used in this study, the uncertainty induced when obtaining $Z_{eff}$ can be eliminated. However, the uncertainty of the elemental component ratio can be reflected in SPR, and the errors for each elemental component are reported in Tables 2 and 3.

## Limitation

In the implementation of our algorithm, we employ a systematic trial-and-error process to identify the optimal elements and their weight fractions for unknown materials. This involves sequentially testing various atomic numbers as potential elements and comparing the outcomes to determine the correct combination of elements with their respective fractional weights. While this approach proves effective for materials with a relatively small number of elements, it may encounter challenges when dealing with complex compounds or mixtures, potentially impacting computational efficiency and restricting its clinical applicability.

A primary concern associated with the trial-and-error process lies in the increased computational burden, particularly evident when materials contain a larger number of elements. As the algorithm considers a growing set of elements, the number of iterations and calculations escalates, placing higher demands on computational resources and leading to extended processing times. Such resource-intensive requirements might hinder its practicality for clinical applications.

Furthermore, the trial-and-error process may encounter difficulties when materials exhibit similar or overlapping elemental compositions. When two or more elements possess comparable mass attenuation coefficients or fractional

weights, accurately distinguishing between them becomes challenging. This ambiguity can prolong convergence times and even result in misidentifications of the elemental composition.

To address these concerns and enhance the algorithm's efficiency and clinical applicability, we propose several potential strategies. Firstly, search space refinement involves focusing on a subset of elements more likely to be present in the unknown material, leveraging prior knowledge about common materials encountered in radiation therapy or specific medical scenarios. This approach reduces the number of iterations required for convergence and enables faster identification of chemical constituents. Secondly, parallel processing and hardware optimization can be achieved by optimizing the algorithm for parallel processing using GPUs (Graphics Processing Units). This allows multiple computations to be performed simultaneously, significantly reducing processing time, particularly when dealing with large datasets or materials with numerous elements. Lastly, rigorous clinical validation with various materials and compounds is essential to assess the algorithm's accuracy, precision, and robustness in real-world medical scenarios. Feedback from clinical validation will facilitate algorithm refinement and address limitations or challenges observed during testing, ensuring its effective performance in handling complex scenarios and meeting clinical application demands.

Through search space refinement, parallel processing, hardware optimization, and rigorous clinical validation, our algorithm can be optimized for clinical applications. These strategies collectively contribute to improving computational efficiency and accuracy in identifying chemical constituents, ultimately enabling more precise dose calculations and enhancing the overall efficacy of particle therapy.

## Conclusion

We developed an algorithm to obtain the types of elements and their weight fractions using the MACs of dual-energy X rays. Although the algorithm has uncertainty, it succeeded in finding fractional weights and elements of materials consisting of up to 6 constituents and it was found that further study to improve uncertainty was needed. Moreover, we evaluated its feasibility of application to the clinic when DECT is in use.

## Supporting information

**S1 File. Representative energy calculation methodology.** Process for obtaining representative energy from 140 kVp spectral energy for $H_2O$ using SpekCalc and Geant4 simulation.
(DOCX)

## Author contributions

**Conceptualization:** Dong Hyeok Choi, So Hyun Ahn, Kwangwoo Park.

**Formal analysis:** Dong Hyeok Choi, So Hyun Ahn, Kwangwoo Park, Min Cheol Han, Jin Sung Kim.

**Resources:** So Hyun Ahn, Kwangwoo Park, Jin Sung Kim.

**Writing – original draft:** Dong Hyeok Choi.

**Writing – review & editing:** Dong Hyeok Choi, So Hyun Ahn, Kwangwoo Park, Min Cheol Han, Jin Sung Kim.

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
