## [Decision Letter · Decision Letter 0]

PONE-D-24-42552Algorithm for Using Dual Energy Computed Tomography to Determine Chemical Composition: A Feasibility StudyPLOS ONE

Dear Dr. Ahn,

Thank you for submitting your manuscript to PLOS ONE. After careful consideration, we feel that it has merit but does not fully meet PLOS ONE’s publication criteria as it currently stands. Therefore, we invite you to submit a revised version of the manuscript that addresses the points raised during the review process.

We look forward to receiving your revised manuscript.

Kind regards,

Cebastien Joel Guembou Shouop, Ph.D., ME, MS

Academic Editor

PLOS ONE

Journal Requirements:

3. Thank you for stating the following in your Competing Interests section: The authors declare that they have no known competing financial interests or personal relationships that could have appeared to influence the work reported in this paper.

Additional Editor Comments:

In order to process your paper (please note if there are review reports these will be included below), we require:

• A point-by-point response to the comments, including a description of any additional experiments that were carried out and a detailed rebuttal of any criticisms or requested revisions that you disagreed with.

This must be uploaded as a Point-by-point response file. All changes to the manuscript must be highlighted or indicated by using tracked changes.

At this stage, please also ensure you have replaced your initial submission image files with production-quality figures. Figures should not include Figure number labels in the image.

EDITOR AND REVIEWER REPORTS

**Editor comments:**

In this manuscript, Choi et al. report on the use of dual-energy computed tomography (CT) to develop an algorithm to identify the chemical constituents of an unknown material (compound or mixture) and improve the accuracy of material discrimination. Using dual energies 80/140 kVp for spectral CT scans in addition to a set of 50/80 and 80/100 keV for mono-energetic X-rays, their algorithm correctly determined the chemical constituent elements of unknown materials. Their funding was compared to the data provided by the National Institute of Standards and Technology (NIST) and displayed differences of < 0.01% for compounds and 6.02% for mixture in the case of mono-energetic X-rays, and 2.98% for compounds and 6.03% for mixtures for spectral X-rays, respectively

The research has yielded important insights that make it essential for the manuscript to be published in PlosOne. The following reasons support this recommendation:

- Firstly, the research is both original and relevant. It offers valuable information on an important step towards the development of an algorithm for the evaluation of the stopping power ratio (SPR), which is crucial for the calculation of the physical dose of particle radiation in the treatment planning system (TPS). The findings are of great importance to researchers and medical staff involved in CT information in the volume unit. The study can be of great interest if the clinical applicability is demonstrated and concrete outcomes are presented.

- Secondly, the research was conducted with a high level of scientific rigour. Both qualitative and quantitative methods were used to collect and analyze data, enhancing the reliability and validity of the findings. Geant4 results were compared to the existing NIST data base.

**
*General comments:*
**

1. Upon assessing the similarity level of your manuscript, it has been observed that approximately 60% of the content of the main text is derived from your paper posted in the Researchsquare repository (https://doi.org/10.21203/rs.3.rs-2593701/v1). While it is permissible to submit the manuscript to an open archive while it is under review, it is advisable to have the same manuscript submitted to PlosOne in the repository, which would result in an approximate 100% similarity with the file in the repository. This adherence to scientific publication ethics and policy aims to prevent dual submission. Should you have any reasons to disclose to the editorial board of PlosOne regarding this matter, please do so.

2. The title of the manuscript needs to be revised considering changing the expression “a feasibility study”

3. The manuscript's sections need to be renumbered, as there is no numbered section whereas the authors keep referring to sections. E.g. line 209 “mentioned in 2.2.1 and 2.2.2”

4. In addition, some grammatical and sentence structure issues should be addressed during the review process as well as subscript and superscript notations. (e.g. Line 318: “… for the* detection of COVID-19 …”; Line 324: “… regarded as* excellent.”; Line 325: “… the full doses*”

*
**Specific comments:**
*

Title: Need to be revised to convey the main purpose of the study

Abstract: This section is for a large readership and should be written appropriately, avoiding abbreviations not acknowledged by the general public as much as possible (SPR). Consider clearly explaining where its use is unavoidable or define it for the first occurrence, and use it thereafter.

Materials and methods:

Line 103: what do the authors mean by “…when a CT image comprising multiple voxels is applied to the algorithm”

Line 110: “Cylindrical gamma detector: radius of 0.1 mm × height of 0.1 mm ”. Is the description of the detector used in this research realistic and implemented in real experiments?

Lines 114-115: “In this study, CT images based both on full energy spectra simulations and mono-energic beams were simulated. ” How did you simulate CT images?

Figure 1 is confusing. The caption should be revised. The track in Fig. 1 (b) shows that the statistic was too low and the derived data’s uncertainties are expected to be top high. Is the position of the source in (a) the same as in (b)? In (b), the source seems to be emitting from inside the box whereas in (a), it is located at a distance of 10.5 cm.

Lines 142 - 154: the detailed description could be simplified or added as supplementary material as it does not contribute to the research's main objective.

Fig. 2. The last step “<5% difference with input attenuations � output: Final solutions” is confusing. It might lead to an infinite loop if the target of <5% is never reached. Another issue is that you have data with 6%, which questions whether the authors used this algorithm for the data presented in the paper.

Line 168: Make sure all captions appear in the manuscript in the order of their citation in the text. Fig.4 seems to be quoted whereas Fig. 3 is not yet mentioned.

Line 209: “mentioned in 2.2.1 and 2.2.2” ?

Lines 212-215: This description should be enhanced taking into account the update in Fig. 4 caption.

Lines 228-230: “In the case of body tissue materials, the fractional weight of some elements was slightly modified so that the sum of the fractional weights was 100%.” Why did the authors modify the fractional weight of some elements? What was the rationale behind the modification? This is crucial to the end users and readers of the paper.

Results:

Table 2 caption should be revised to consider the content of the table itself.

Line 274: “The reasons for this will be discussed in more detail in section 4.1.”?

Lines 334-336: What is the contribution to the scientific research presented?

Lines 338-339: “Alternatively, we can reduce the number of voxels by skipping the calculation of nearby and similar HU-valued voxels.” What is the direct consequence on the obtained results, if implemented?

Line 407: Is “three to six components” a limitation of the study? What is the rationale for setting this limit?

Conclusions:

Please, summarize the main findings of your research.

**Reviewer Comments:**

**Reviewer 1:**

*
**General comment to the authors:**
*

The study is well-prepared and has significant results on an important topic. But some revisions are required. The specific comments are also given in the .doc file of the manuscript as comments. It is also necessary to correct some technical mistakes in the manuscript (reference citation and full stop before reference citation).

*
**Specific comment:**
*

Lines 35-36: “For mono-energetic X-rays, the differences were < 0.01% for compounds and 6.02% for mixture, respectively,” I think no need «respectively» at the end of the sentence.

Line 111: Precisely, which material did the authors use to fill this box-shaped material in the GEANT4 simulation?

Lines 125-126: the author hereby stated that: “In principle, applying the algorithm for identifying chemical constituent from spectral energy, we should integrate over the complete energy spectrum”. How the energy spectrum was obtained?

Line 127: I suggest changing the word "Mono energy" in the title to "monoenergy”.

Lines 165-166: “However, the noise in the image and uncertainty in measurement, f(z) would not vanish but become close to 0” What are the underlying factors contributing to noise and uncertainty in image data?

Line 195: I suggest to change the word “Utilized” to “used”

Lines 208-209: The authors mentioned “the formulas mentioned in 2.2.1 and 2.2.2” but there are not these formulas in the whole manuscript.

I suggest to change the word "mono energies" to "monoenergies” in all the text

Lines 275-277: “It was confirmed that the %difference in the case of number of constituents was 4, 5 and 6 were 0.340 – 5.449%, 1.410 – 5.188%, and 3.693 – 6.027%, respectively” What does the author intend to convey? If possible, rephrase this sentence for more clarity.

Lines 307-308: “It is considered a useful result, but several considerations were made about the problems encountered during the performing of this study and are described below.” I suggest rephrasing this sentence to enhance the clarity and convey the intended meaning more precisely.

Line 313: I suggest to change the word “quantized” to “quantified”

Line 232: in the phrase «The algorithm was tested on five compounds that combine C, H, O elements...» which compounds? I suggest citing them.

Lines 233-234: “In the case of body tissue materials, the fractional weight of some elements was slightly modified” Which elements? I suggest to enumerate the elements

Line 390-391: The authors state that to acquire both data, 1E+08 events were used for the simulation. To confirm whether data is statistically reliable, the number of events was changed and MACs were obtained ten times for each event” Why the events number was changed? Are there some methods that explain this?

If not, how did the authors go about changing the number of events, given that they defined 108 to obtain the data?

For each event, MACs were obtained ten times. For each defined energy, there must be a MAC value. If the MAC value for an event is obtained ten times then what is the total MAC average (for energy or intensity)?

Line 457: I suggest that the authors start this by “find the space …”

**Reviewer 2:**

I am pleased to evaluate the manuscript titled "Algorithm for Using Dual Energy Computed Tomography to Determine Chemical Composition: A Feasibility Study." Below are my comments on this work:

The study addresses an important gap by proposing an algorithm to determine chemical composition using dual-energy CT, advancing beyond the typical use of effective atomic numbers (Zeff). This approach is potentially valuable as it could reduce uncertainty in SPR (Stopping Power Ratio) calculations, which is essential for improving dose accuracy in radiation therapy.

The authors effectively strengthen their findings by employing both NIST data and Geant4 simulations for mass attenuation coefficient (MAC) values, enhancing the reliability of the results.

The manuscript is logically structured with clear sections for the introduction, methodology, results, and conclusion. Each part is well-organized, and figures and tables, particularly those comparing simulation results with NIST data, are presented effectively.

Although the article briefly addresses the impact of noise in CT images, a more detailed discussion on this topic could strengthen the study’s applicability, as noise can affect MAC calculations and the algorithm’s accuracy in clinical settings.

The work could benefit from clearer language and an expanded discussion on noise resilience. Nevertheless, the study remains scientifically sound, with promising potential for practical application.

In summary, I recommend that the authors consider these suggestions to enhance readability, and robustness, which would make this study a more comprehensive and impactful contribution to medical physics.

Reviewers' comments:

Reviewer's Responses to Questions

**Comments to the Author**

1. Is the manuscript technically sound, and do the data support the conclusions?

Reviewer #1: Yes

Reviewer #2: Yes

2. Has the statistical analysis been performed appropriately and rigorously? 

Reviewer #1: Yes

Reviewer #2: Yes

3. Have the authors made all data underlying the findings in their manuscript fully available?

Reviewer #1: Yes

Reviewer #2: Yes

4. Is the manuscript presented in an intelligible fashion and written in standard English?

Reviewer #1: Yes

Reviewer #2: Yes

5. Review Comments to the Author

Reviewer #1: PLOS ONE

Manuscript Number: PONE-D-24-42552

Full Title: Algorithm for Using Dual Energy Computed Tomography to Determine Chemical Composition: A Feasibility Study

General comment to the author’s:

The study is well-prepared and has significant results on an important topic. But some revisions are required. The specific comments are also given in the .doc file of the manuscript as comments. It is also necessary to correct some technical mistakes in the manuscript (reference citation and full stop before reference citation).

Specific comment:

Lines 35-36: “For mono-energetic X-rays, the differences were < 0.01% for compounds and 6.02% for mixture, respectively” I think no need «respectively» at the end of the sentence.

Line 111: Precisely, which material the authors are used to fill this Box shaped material in GEANT4 simulation?

Lines 125-126: the author hereby stated that: “In principle, applying the algorithm for identifying chemical constituent from spectral energy, we should integrate over the complete energy spectrum”.How the energy spectrum was obtained?

Line 127: I suggest to change the word "Mono energy" in the title to "monoenergy”.

Lines 165-166: “However, the noise in the image and uncertainty in measurement, f(z) would not vanish but become close to 0” What are the underlying factors contributing to noise and uncertainty in image data?

Line 195: I suggest to change the word “Utilized” to “used”

Lines 208-209: The authors mentioned “the formulas mentioned in 2.2.1 and 2.2.2” but there are not these formulas in all the manuscript.

I suggest to change the word "mono energies" to "monoenergies” in all the text

Lines 275-277: “It was confirmed that the %difference in the case of number of constituents was 4, 5 and 6 were 0.340 – 5.449%, 1.410 – 5.188%, and 3.693 – 6.027%, respectively” What does the author intend to convey? If possible, rephrase this sentence for more clarity.

Lines 307-308: “it is considered a useful result, but several considerations were made about the problems encountered during performing of this study and are described below.” I suggest to rephrase this sentence to enhance the clarity and convey the intended meaning more precisely.

Line 313: I suggest to change the word “quantized” to “quantified”

Line 232: in the phrase «The algorithm was tested on five compounds that combine C, H, O elements...» which compounds? I suggest to cite them.

Lines 233-234: “In the case of body tissue materials, the fractional weight of some elements was slightly modified” which elements? I suggest to enumerate the elements

Line 390-391: The authors states that to acquire both data, 1E+08 events were used for the simulation. To confirm whether data is statistically reliable, the number of events was changed and MACs was obtained ten times for each event” Why the events number was changed? There are some methods that explain this?

If not, how did the authors go about changing the number of events, given that they defined 108 to obtain the data?

For each event, MACs were obtained ten times. For each defined energy, there must be a MAC value. If the MAC value for an event is obtained ten times then total MAC is the average (for an energy or intensity)?

Line 457: I suggest that the authors start this by “find the space …”

Reviewer #2: I am pleased to evaluate the manuscript titled "Algorithm for Using Dual Energy Computed Tomography to Determine Chemical Composition: A Feasibility Study." Below are my comments on this work:

The study addresses an important gap by proposing an algorithm to determine chemical composition using dual-energy CT, advancing beyond the typical use of effective atomic number (Zeff). This approach is potentially valuable as it could reduce uncertainty in SPR (Stopping Power Ratio) calculations, which is essential for improving dose accuracy in radiation therapy.

The authors effectively strengthen their findings by employing both NIST data and Geant4 simulations for mass attenuation coefficient (MAC) values, enhancing the reliability of the results.

The manuscript is logically structured with clear sections for the introduction, methodology, results, and conclusion. Each part is well-organized, and figures and tables, particularly those comparing simulation results with NIST data, are presented effectively.

Although the article briefly addresses the impact of noise in CT images, a more detailed discussion on this topic could strengthen the study’s applicability, as noise can affect MAC calculations and the algorithm’s accuracy in clinical settings.

The work could benefit from clearer language and an expanded discussion on noise resilience. Nevertheless, the study remains scientifically sound, with promising potential for practical application.

In summary, I recommend that the authors consider these suggestions to enhance readability, robustness, which would make this study a more comprehensive and impactful contribution to medical physics.

6. PLOS authors have the option to publish the peer review history of their article (what does this mean? ). If published, this will include your full peer review and any attached files.

**Do you want your identity to be public for this peer review?** For information about this choice, including consent withdrawal, please see our Privacy Policy .

Reviewer #1: No

Reviewer #2: No

---

## [Author Response · Author response to Decision Letter 1]

13 Mar 2025

PLOS ONE

Manuscript Number: PONE-D-24-42552

Full Title: Algorithm for Using Dual Energy Computed Tomography to Determine Chemical Composition: A Feasibility Study

General comment to the author’s:

The study is well-prepared and has significant results on an important topic. But some revisions are required. The specific comments are also given in the .doc file of the manuscript as comments. It is also necessary to correct some technical mistakes in the manuscript (reference citation and full stop before reference citation).

Specific comment:

Lines 35-36: “For mono-energetic X-rays, the differences were < 0.01% for compounds and 6.02% for mixture, respectively” I think no need «respectively» at the end of the sentence.

The manuscript has been revised according to your suggestion.

L35: For mono-energetic X-rays, the differences were < 0.01% for compounds and 6.02% for mixture.

Line 111: Precisely, which material the authors are used to fill this Box shaped material in GEANT4 simulation?

Thank you for your comment. In this study, box-shaped materials were individually modeled for elements with atomic numbers ranging from 1 to 20 in the Geant4 simulation. This approach was employed to evaluate the X-ray attenuation characteristics of different elements.

We will clarify this information in the manuscript to ensure better understanding.

Sentence to be added in the manuscript:

L113: In the Geant4 simulation, each box-shaped material was filled with a single element corresponding to atomic numbers ranging from 1 to 20 to obtain MACs data for individual elements.

Lines 125-126: the author hereby stated that: “In principle, applying the algorithm for identifying chemical constituent from spectral energy, we should integrate over the complete energy spectrum”.How the energy spectrum was obtained?

Thank you for your insightful question. In this statement, we were referring to the theoretical approach of integrating over the complete energy spectrum to identify chemical constituents.

To validate the selection of representative energy, we compared the MACs obtained from the full energy spectrum with those derived from specific monoenergies representing the spectral X-ray. The representative energy was determined based on the condition that the MAC of a specific reference material (H₂O) matched the MAC integrated over the full energy spectrum. This method was further validated using PMMA and cortical bone (ICRP) to ensure consistency across different tissue-equivalent materials.

For further details on how the representative energy for H₂O, PMMA, and cortical bone was determined, please refer to the later sections of the manuscript and supplement document.

Added Sentence

L139: To provide a detailed explanation of the process for obtaining representative energy, a supplementary document has been included. This document outlines the step-by-step methodology used to determine the representative energy for 140 kVp spectral energy, specifically for H₂O, using SpekCalc, Geant4 simulations, and NIST data interpolation.

Line 127: I suggest to change the word "Monoenergy" in the title to "monoenergy”.

We have revised the word by changing "Monoenergy" to "monoenergy" as suggested.

Lines 165-166: “However, the noise in the image and uncertainty in measurement, f(z) would not vanish but become close to 0” What are the underlying factors contributing to noise and uncertainty in image data?

Thank you for your comment. Considering the possibility that the explanation of noise and measurement uncertainty might not be sufficiently clear, we have added a sentence indicating that this topic is discussed in detail in the Discussion section. This revision ensures that readers can better understand the discussion regarding noise and measurement uncertainty.

Added sentence:

L174: The factors contributing to noise in the image and measurement uncertainty are discussed in the Discussion section.

The main factors contributing to noise and uncertainty in image data in this study are as follows:

Noise and uncertainty in CT scanning

Evaluation of noise impact: In this study, we analyzed the effect of CT image noise on the performance of our algorithm. Hünemohr et al. evaluated weight fraction estimation by incorporating Gaussian-distributed noise into electron density and the effective atomic number (Zeff), demonstrating prediction errors across different elements (Discussion, Evaluating the effect of noise section).

Mitigating noise effects using PCA: Lalonde and Bouchard proposed using a larger number of energy spectra than principal components (PC) in principal component analysis (PCA) to mitigate the effect of noise. In our study, we also assessed whether the developed algorithm could accurately determine elemental composition in noisy conditions (Discussion, Evaluating the effect of noise section).

Uncertainty in mass attenuation coefficients (MACs)

The MAC is derived from HU values obtained via CT scans, and noise and measurement errors can be introduced in this process. According to the International Atomic Energy Agency (IAEA), CT values should remain within ±20 HU of the manufacturer’s recommended values, leading to approximately 2% uncertainty in MACs relative to water as the reference (Discussion, Evaluating the effect of noise section).

To ensure a conservative assessment, we set an upper error limit of 3% in our experiments. If an error exceeding 3% is introduced into the MAC, the fractional weight calculated by our algorithm exhibited an additional error exceeding 5% (Discussion, Evaluating the effect of noise section).

Statistical error in Geant4 simulations

The built-in and input data for the algorithm were obtained through Geant4 simulations. To assess statistical reliability, we varied the number of events and calculated MAC values multiple times. The results showed that with 1E+08 events, the standard deviation was 5.68E-05, and the relative error remained below 0.01, confirming the statistical reliability of the simulation (Discussion, Statistical error of Geant4 simulation section).

These factors contribute to noise and uncertainty in image data, and our study has taken them into account to evaluate the reliability of the proposed algorithm.

Line 195: I suggest to change the word “Utilized” to “used”

We have revised the text by changing "Utilized" to "used" as suggested.

Lines 208-209: The authors mentioned “the formulas mentioned in 2.2.1 and 2.2.2” but there are not these formulas in all the manuscript.

Thank you for your comment. We have identified that referring to "the formulas mentioned in 2.2.1 and 2.2.2" could be misleading since these specific section numbers do not exist in the manuscript. To enhance clarity, we have revised the phrase to "the formulas mentioned in the Theory section eq 1 to 6." This ensures that readers can properly refer to the relevant theoretical background.

L219: As an example of identifying the atomic number of an unknown material through the formulas mentioned in the Theory section eq 1 to 6.

Lines 275-277: “It was confirmed that the %difference in the case of number of constituents was 4, 5 and 6 were 0.340 – 5.449%, 1.410 – 5.188%, and 3.693 – 6.027%, respectively” What does the author intend to convey? If possible, rephrase this sentence for more clarity.

Thank you for your comment. We have revised the sentence to clarify the range of %difference for different numbers of constituents in the mixtures. This modification ensures that readers can better understand the presented data and its implications.

Revised Sentence

L296: The results in Table 3 show that the %difference ranged from 0.340% to 5.449% for mixtures with four constituents, 1.410% to 5.188% for five constituents, and 3.693% to 6.027% for six constituents. This indicates that as the number of constituents increases, the overall %difference tends to increase. The reasons for this will be discussed in the Discussion section.

Lines 307-308: “it is considered a useful result, but several considerations were made about the problems encountered during performing of this study and are described below.” I suggest to rephrase this sentence to enhance the clarity and convey the intended meaning more precisely.

Thank you for your comment. We have revised the sentence to improve clarity and ensure that the intended meaning is conveyed more precisely.

Revised Sentence

L323: The originality of our algorithm lies in its ability to determine the constituent elements and their fractional weights for an unknown material containing two or more elements, using only two CT energy spectra. Unlike conventional approaches that approximate material composition using Zeff, our method directly estimates elemental fractions. However, challenges remain in accurately interpreting CT images due to inherent noise and measurement uncertainties. In particular, image noise and variations in CT number can affect the reliability of elemental composition analysis.

Line 313: I suggest to change the word “quantized” to “quantified”

Thank you for your suggestion. We have revised the text by changing "quantized" to "quantified" as recommended.

Line 232: in the phrase «The algorithm was tested on five compounds that combine C, H, O elements...» which compounds? I suggest to cite them.

Thank you for your valuable comment. We have revised the text to specify the five compounds used in our evaluation: H₂O (water), C₂H₄ (ethylene), C₈H₈ (styrene), C₅O₂H₈ (valeric acid), and C₁₀H₈O₄ (naphthalic acid).

Revised Sentence

L329: The algorithm was tested on five compounds—H₂O (water), C₂H₄ (ethylene), C₈H₈ (styrene), C₅O₂H₈ (valeric acid), and C₁₀H₈O₄ (naphthalic acid)—which contain C, H, and O elements, as well as nine mixtures of the body tissue materials mentioned [18, 19].

Lines 233-234: “In the case of body tissue materials, the fractional weight of some elements was slightly modified” which elements? I suggest to enumerate the elements

Thank you for your valuable comment. We have revised the text to specify the elements whose fractional weights were modified. In the case of body tissue materials, the fractional weight of hydrogen (H) and carbon (C) was slightly adjusted to ensure that the sum of all fractional weights equaled 100%.

Revised Sentence

L241: In the case of body tissue materials, the fractional weight of hydrogen (H) and carbon (C) was slightly modified to ensure that the total fractional weight summed to 100%.

Line 390-391: The authors states that to acquire both data, 1E+08 events were used for the simulation. To confirm whether data is statistically reliable, the number of events was changed and MACs was obtained ten times for each event” Why the events number was changed? There are some methods that explain this?

Thank you for your question. The number of events was varied to evaluate the statistical reliability of the Monte Carlo simulation results. In Monte Carlo-based simulations, statistical fluctuations are inherent, and increasing the number of events generally reduces these fluctuations, improving the precision of estimated parameters such as the Mass Attenuation Coefficients (MACs). To ensure that our results were not significantly affected by statistical noise, we systematically changed the number of events and analyzed how it impacted the computed MAC values. This approach aligns with standard practices in Monte Carlo simulations for assessing convergence and stability.

If not, how did the authors go about changing the number of events, given that they defined 108 to obtain the data?

The number of events was modified incrementally by setting different predefined values (1E+03, 1E+04, 1E+05, 1E+06, 1E+07, 5E+07, and 1E+08) and running the simulation separately for each case. This allowed us to observe the impact of event count on MAC stability. The results from each event configuration were compared to determine the minimum number of events required to achieve statistical reliability. This method follows common Monte Carlo validation techniques used in radiation transport simulations.

For each event, MACs were obtained ten times. For each defined energy, there must be a MAC value. If the MAC value for an event is obtained ten times then total MAC is the average (for an energy or intensity)?

Yes, for each event configuration, MAC values were obtained ten times, and for each defined energy level, an individual MAC value was computed. The final MAC value for a given energy was calculated as the average of these ten values. This averaging process reduces statistical fluctuations and provides a more stable estimate of the MAC for each energy level. The standard deviation and relative error of these ten values were also analyzed to quantify the uncertainty and ensure the robustness of our results.

Line 457: I suggest that the authors start this by “find the space …”

Thank you for your suggestion. We have revised the sentence to begin with "Find the space," as recommended. This modification enhances clarity and aligns with the intended meaning of the section.

Revised Sentence

L483: Firstly, find the space of relevant elements by refining the search space to focus on a subset of elements more likely to be present in the unknown material ~

Editor comments:

In this manuscript, Choi et al. report on the use of dual-energy computed tomography (CT) to develop an algorithm to identify the chemical constituents of an unknown material (compound or mixture) and improve the accuracy of material discrimination. Using dual energies 80/140 kVp for spectral CT scans in addition to a set of 50/80 and 80/100 keV for mono-energetic X-rays, their algorithm correctly determined the chemical constituent elements of unknown materials. Their funding was compared to the data provided by the National Institute of Standards and Technology (NIST) and displayed differences of < 0.01% for compounds and 6.02% for mixture in the case of mono-energetic X-rays, and 2.98% for compounds and 6.03% for mixtures for spectral X-rays, respectively

The research has yielded important insights that make it essential for the manuscript to be published in PlosOne. The following reasons support this recommendation:

- Firstly, the research is both original and relevant. It offers valuable information on an important step towards the development of an algorithm for the evaluation of the stopping power ratio (SPR), which is crucial for the calculation of the physical dose of particle radiation in the treatment planning system (TPS). The findings are of great importance to researchers and medical staff involved in CT information in the volume unit. The study can be of great interest if the clinical applicability is demonstrated and concrete outcomes are presented.

- Secondly, the research was conducted with a high level of scientific rigour. Both qualitative and quantitative methods were used to collect and analyze data, enhancing the reliability and validity of the findings. Geant4 results were compared to the existing NIST data base.

General comments:

1. Upon assessing the similarity level of your manuscript, it has been observed that approximately 60% of the content of the main text is derived from your paper posted in the Researchsquare repository (https://doi.org/10.21203/rs.3.rs-2593701/v1). While it is permissible to submit the manuscript to an open archive while it is under review, it is advisable to have the same manuscript submitted to PlosOne in the repository, which would result in an approximate 100% similarity with the file in the repository. This adherence to scientific publication ethics and policy aims to prevent dual submission. Should you have any reasons to disclose to the editorial board of PlosOne regarding this matter, please do so.

We would like to clarify that this manuscript has not been submitted to any other journal apart from PLOS ONE, and there is no case of dual submission.

The paper posted in the ResearchSquare repository is a preprint version, which was uploaded to share our research findings with the academic community and facilitate discussions. This aligns with PLOS ONE's preprint policy and follows the appropriate submission process.

The preprint version remains un

---

## [Decision Letter · Decision Letter 1]

Algorithm for Using Dual Energy Computed Tomography to Determine Chemical Composition: Method Development and Validation

PONE-D-24-42552R1

Dear Dr. Ahn,

We’re pleased to inform you that your manuscript has been judged scientifically suitable for publication and will be formally accepted for publication once it meets all outstanding technical requirements.

Kind regards,

Cebastien Joel Guembou Shouop, Ph.D., ME, MS

Academic Editor

PLOS ONE

Additional Editor Comments (optional):

Overall, the author did a great job in revising the manuscript.

One thing left to the author is whether the state of the five compounds tested can be generalized to other types of materials (perhaps H2O in solid (ice), liquid (water), and gas (vapor) states).

Reviewers' comments:

Reviewer's Responses to Questions

**Comments to the Author**

1. If the authors have adequately addressed your comments raised in a previous round of review and you feel that this manuscript is now acceptable for publication, you may indicate that here to bypass the “Comments to the Author” section, enter your conflict of interest statement in the “Confidential to Editor” section, and submit your "Accept" recommendation.

Reviewer #1: All comments have been addressed

2. Is the manuscript technically sound, and do the data support the conclusions?

Reviewer #1: Yes

3. Has the statistical analysis been performed appropriately and rigorously? 

Reviewer #1: Yes

4. Have the authors made all data underlying the findings in their manuscript fully available?

Reviewer #1: Yes

5. Is the manuscript presented in an intelligible fashion and written in standard English?

Reviewer #1: Yes

6. Review Comments to the Author

Reviewer #1: (No Response)

7. PLOS authors have the option to publish the peer review history of their article (what does this mean? ). If published, this will include your full peer review and any attached files.

**Do you want your identity to be public for this peer review?** For information about this choice, including consent withdrawal, please see our Privacy Policy .

Reviewer #1: No

---

## [Editor Report · Acceptance letter]

PONE-D-24-42552R1

PLOS ONE

Dear Dr. Ahn,

I'm pleased to inform you that your manuscript has been deemed suitable for publication in PLOS ONE. Congratulations! Your manuscript is now being handed over to our production team.

Kind regards,

on behalf of

Dr. Cebastien Joel Guembou Shouop

Academic Editor

PLOS ONE